# Phygital as a Lever for Value Propositions in Italian Cultural Tourism Startups

**Fabio Greco** [1,*] , **Francesco Carignani** [2] , **Laura Clemente** [3] **and Francesco Bifulco** [1]

1. Department of Humanities, University of Naples Federico II, 80131 Naples, Italy; francesco.bifulco@unina.it
2. Economics, Management and Institutions, University of Naples Federico II, 80131 Naples, Italy; francesco.carignani@unina.it
3. Department of Sciences of Antiquity, Sapienza University of Rome, 00185 Rome, Italy; laura.clemente@uniroma1.it
* Correspondence: fabio.greco@unina.it

**Abstract:** The proliferation of innovative digital technologies is increasingly evident in the domains of culture and tourism. This trend, characterized by significant a potential for experimentation and practical application, necessitates a comprehensive understanding of the emerging tools that are enhancing the cultural tourism sector. Central to this evolution is the emergence of cultural startups that are leveraging advanced technological solutions to revolutionize market dynamics. In the wake of the COVID-19 pandemic, which severely impacted the cultural sector, there is a pressing need for innovation. This study combines the concept of cultural tourism startups with the notion of 'phygital'—a blend of physical and digital realities—aiming to augment the existing, yet limited, body of research in this field. This research seeks to provide insights into the market trends shaped by cultural startups, focusing on tourism. By examining case studies of Italian cultural tourism startups that are implementing innovative and engaging 'phygital' strategies, we aim to offer theoretical contributions to the discourse on phygital applications in culture, as well as practical recommendations for the managers and founders of cultural tourism startups that are venturing into new business models. The selected startups are primarily engaged in enhancing the experiences of incoming tourists, improving customer and partner interactions, and promoting the conservation of Italy's cultural heritage.

**Keywords:** tourism startups; cultural startups; phygital technology; value proposition

## 1. Introduction

Digital transformation has generated radical changes in everyday life and pervasive interconnections between people and businesses, with different implications in terms of customer engagement and the generation of new value propositions [1]. The ability of digital technologies to transcend physical environments reflects an unprecedented architecture in the interactions between a value-creating enterprise and its intended recipients, and it is into this paradigm that the phygital approach fits.

The term phygital (synthesis between the words 'physical' and 'digital') was originally used to describe the connections between the physical and digital worlds [2]. Over the years, numerous publications have focused on discovering the theoretical foundations and managerial applications of phygital spaces [3,4], and there are many facets to which this conceptualization is open.

Interestingly, if a set of technologies in continuous and rapid evolution is constantly in search of 'content' to be processed and conveyed to infinite audiences of users, in particular to the social media audience, it is equally interesting to highlight how one of the sectors that offers more food for thought in this sense is the cultural tourism sector, a privileged field of study on the production of content created for an audience willing to receive and enjoy it. An example is the approximately 4.4 million downloads registered for the video game

application 'Father and Son' created by the National Archaeological Museum of Naples (Sole24ore), which contributed to the definition of a new model of interaction and learning for tourists. Analysing these dynamics, it is evident that the cultural tourism sector is experiencing a renaissance in the post-COVID 19 era thanks to the implementation of new technologies and the development of value propositions in line with the changing needs of all stakeholders involved [5].

This research aims to investigate innovative companies that use this approach, with particular reference to startups, focusing this research area in the context of cultural tourism.

The startup phenomenon, which continues to grow worldwide, sees about 14,000 startups registered in the special section of the business register in Italy [6], with an average turnover of EUR 167,000 and a total of almost 60,000 employees. Of these, 37.7 percent belong to the cultural and creative sector [7]. In Italy, the National Plan for Recovery and Resilience oversees the allocation of numerous economic resources, amounting to EUR 6.675 billion, with the aim of increasing the level of attractiveness, through digitization, of museum collections, archives and libraries, and cultural venues, with obvious implications for the cultural tourism sector.

This deployment of economic resources aimed at revitalizing culture-related tourism activities in the post-COVID 19 era in sustainable ways, and the emergence of the startup phenomenon together with the development of innovative private initiatives in the business of culture, make it useful to investigate the modus operandi of these new realities in the sector and their implementation of new technologies.

RQ: How does the phygital formula drive value propositions in Italian cultural tourism startups?

This question aims to explore the intersection of technology, culture, and tourism through the lens of innovative Italian startups, investigating the implementation of phygital strategies in these interrelated sectors.

The main lines of research concerning the adoption of the phygital approach in the context of cultural startups are illustrated below, followed by the research design and the analysis methodology adopted.

## 2. Literature Review

At the vanguard of digital transformation, the 'phygital' paradigm emerges as a synthesis of tactile and virtual information, a concept that combines the latest communication technologies with tangible environments. This integrative model is increasingly recognized within scholarly discourse for its transformative impact on business models [8] and consumer engagement [9]. In fact, in recent decades, in the tourism sector, digitalisation and the proliferation of mobile devices have led to the creation of real online tourist communities, where, from the perspective of a travel experience, the factor of immediacy in sharing information is essential [10], as is the immediacy of being able to use real-time services while traveling, with the alternative being losing the customer's attention and preference [11]. Contemporary research delineates the phygital experience through the lenses of immediacy, immersion, and interaction, advocating for a seamless blend of the physical and digital realms to create value-added consumer experiences [12,13]. This convergence catalyses economic growth by leveraging digital platforms to foster a vibrant ecosystem in which interactivity is the cornerstone [14]. Furthermore, the increase in the Internet of Things (IoT) underscores the dissolution of the boundaries between the actual and the virtual, creating a world where these realms not only intersect but are also mutually transformative [15–17].

In dissecting the phygital approach, marketers identify three salient features—immediacy, immersion, and interaction—heralding a new epoch in the provisioning of goods and services, where personalization and emotional connection are paramount. Technological spaces such as augmented reality, mixed reality, and extended reality are instrumental to this paradigm, suggesting an imperative for perpetual innovation in customer engagement [18–20]. This literature review synthesizes the aforementioned insights to unpack the triadic relationship between cultural startups, the phygital concept, and their interplay,

highlighting the imperative for cultural and tourism startups to adapt to this phygital narrative. The emergent literature posits the phygital approach as a crucible for community and value creation [21], necessitating a reimagined value proposition that transcends product centricity to focus on experiential differentiation [22,23].

This introduction thus sets the stage for a deeper exploration of the literature, establishing a framework that critically examines the dynamic interplay between cultural startups and the phygital paradigm and how it informs their evolving value propositions within the cultural and tourism sectors.

### 2.1. Value Proposition using the Phygital Formula

The concept of the phygital concerns the most up-to-date knowledge and innovations in the field of communication technologies, combined with the physical environment. This concept is gaining increasing attention within the international scientific community [9], so much so that this phenomenon is also focusing attention on new digital business models [8]. In fact, in recent decades, in the tourism sector, digitalisation and the proliferation of mobile devices have led to the creation of real online tourist communities, where, from the perspective of a travel experience, the factor of immediacy in sharing information is essential [10], as is the immediacy of being able to use real-time services while traveling, with the alternative being losing the customer's attention and preference [11]. The strong internationalization of tourism has thus influenced the increase in the degree of innovation [24], a factor which, together with the development of new products and services, has influenced the creation of new value for visitors, but also the development of sustainability for businesses [25]. So over the years, the literature has therefore outlined characteristics such as immediacy, immersion, and interaction, which characterize these innovative experiences when purchasing products and services (Taheri et al., 2019), thus leading to the exploitation of these potentials thanks to the true added value that the physical and digital levels together bring to the consumer, consolidating the connections between digital experiences and physical improvements [13]. It is precisely this strong interactivity of the ecosystem, through the use of digital platforms, that generates growth and important economic value [14]. In fact, the rapid rise of the IoT [16] represents a turning point in the contrast between the real and virtual world, showing them not as distinct and separate areas, but as areas which can overlap [15] or replace each other [17]. In fact, according to the literature and the operators who use phygital-related technologies, there is an intrinsic connection between digital functions and the customer's physical experience, which takes on the nuances of a bond, which becomes inseparable, between the physical plane and the digital one [26].

The literature that focuses specifically on marketing identifies three main aspects in the field of the phygital, identified as the three "i"s: immediacy (the offer of immediate goods and services), immersion (developing the brand experience around the user), and interaction (developing an emotional relationship in the purchasing process).

The modalities through which the phygital approach can manifest itself are different, such as augmented reality [18], mixed reality [19], and extended reality [20]. With respect to these actions, through which companies interact with their customer targets, it appears necessary to make the combination of these channels continuous, in order to guarantee the customer the best possible experience, and also to provide the company that provides these products or services with detailed information on the needs that are coming from outside [27]. Within this scenario, characterized by the immediacy of the response to the needs of the user community, companies are called upon to design their products and services using this mixed approach, which is aimed at preparing for new opportunities and predicting customer involvement, as reported [28]. It is therefore precisely this phygital approach that allows for the formation of new communities capable of participating in the creation of value [21].

As relationships with the customer change, especially through social media, the phygital approach must take into account large changes in value proposition [23]. The

relationship with users/customers changes as collaboration processes are initiated with companies [29]. The purchasing process changes, which, taken to a level beyond the physical, includes a very strong emotional aspect, capable of leading to much higher levels of engagement [30]. Here, in some fields, the use of AI can have a very positive impact on the value proposition of phygital-related services [31]. But, in the company's relationship with the customer, the value proposition also changes due to this intrinsic relationship: the relationship transforms into an immediate one, which allows the company to offer a non-standardized experience, but one that is extremely different from any other. Here, where the customer has an active role in customization [32], these products and services are made more inclusive [33]. All these changes lead to a paradigm shift, where the value proposition is no longer focused on the product itself, but on the experience surrounding the acquisition of that product [22].

### 2.2. Tourism- and Culture-Based Startups

The ecosystem of cultural startups that are developing value propositions in the tourism sector is complex and full of variables; these variables are as important as the ecosystems of startups in general [34]. Cultural tourism startups are key drivers of the evolution of the tourism market and significantly contribute to the spread of innovation. By focusing on cultural aspects, these startups facilitate the dissemination of new ideas and practices and business models [35].

Cultural startups are unique in the business landscape, primarily for two reasons.

The first is their "interdisciplinarity" [36], which signifies their adeptness at integrating creative talent with business and management expertise.

Second, they exhibit "limited scalability," indicating that investments in this sector are often modest. This is attributed to the challenges in replicating their services across various regions and scaling up their revenues without proportionately increasing their costs and management resources [37].

Zaman's study [38] delves into the dynamics of cultural heritage entrepreneurship (CHE) in countries participating in a research project focused on CHE's role in economic recovery and sustainable development. The research addresses various challenges that CHE faces, such as limited financial resources, difficulties in market access, innovation barriers, issues surrounding intellectual property rights, and the need for specialized education and training. The study underscores the fact that cultural entrepreneurship plays a pivotal role in the market through the provision of goods and services, and by organizing and managing cultural heritage in either a commercial or nonprofit manner. This approach is determined by the unique nature of cultural heritage, and whether it is viewed as an asset, a form of cultural capital, or a pure public good.

Another interesting area of study, for the purposes of this research, is that which compares the concept of tourism to that of the business of culture. The author posits that activities impacting cultural heritage, such as archaeology, are part of a series of processes that transmit culture to society. Specifically, the author argues that archaeology plays a crucial role in telling the stories of different people, thereby fostering tourism by drawing on the historical narratives unearthed by archaeological endeavours.

In Bertasini's 2020 study [35], an analysis of the global distribution of cultural startups, including those in the tourism, cultural tourism, and art sectors, revealed a notable concentration of cultural entrepreneurship. North America and Europe led with 38.7% and 38.5%, respectively, of such startups. In particular, European startups experienced steady and significant growth from 2013 to 2019. In contrast, Asian startups, constituting 20.4%, experienced notable expansion until 2015, followed by a downwards trend. The study also highlights that a predominant focus of these startups, amounting to 62.5% of them, is on booking and purchasing services such as tours and local experiences. The remaining 17.5% primarily concentrate on enhancing consumer experiences. There are few studies concerning the ways in which cultural organizations are influenced by digitization [39,40].

Instead, research analysing cultural digital startups that enhance their core business with the support of new technologies has flourished [41].

Through the generation of smart solutions through the application of new technologies that are defined as multimedia content, cultural entrepreneurs provide visitors with an enriched, interactive, and easily accessible visiting experience.

### 2.3. Phygital Integration into Cultural Tourism Startups

If, in general, new technologies are seen as an indispensable tool for new startups and their scalability [34], then, in the tourism sector, the use of new technologies may even be considered the main tool for structuring their offers, as well as a fundamental element of a dynamic approach to customers [42], when considering the increase in the competitiveness of a destination which exploits technologies to promote itself [43]. Due to new technologies, the experience of a tourist space has taken on a hybrid dimension, in which the information obtained in the virtual environment now enriches all phases of the customer journey, thus making it so they use the latest technologies; the use of new strategies must take into account a holistic approach [44].

In the cultural tourism sector, COVID-19 represented an interesting case for analysis due to the impact it had on digitalisation in the museum sector and the digitalisation choices of museum organizations [45]. But a positive impact has also been had with other cultural tourist sites, where the use of digital technology has made it possible to better regulate files and access in line with COVID directives, but also offering more engaging visit experiences [46]. This experience invites reflection on the new normality for cultural sites, where the phygital experience, the on-demand experience, and the more inclusive experience take on particular relevance, especially with reference to a millennial audience [47]. Furthermore, it should not be overlooked how the phygital can, by modifying the value proposed to the user, open the doors to digital ecosystems, which represent an aspect with high sustainability for cultural enterprises [48]. The phygital could represent an important attraction in the cultural sector for, for example, an audience of "millennials" who, during the entire customer journey, activate online and offline interactions that make the limits between the physical and digital world less defined [49]. The overlap between the physical and digital also translates into the integration of the visit experience with innovative and high-tech supports, such as interactive multimedia content, virtual reality, augmented reality, and geolocation services. In fact, augmented reality and virtual reality, specifically, are considered among the phygital experiences with the highest degree of functional and emotional participation for customers [50]. The use of AR to complement the tourist visit experience has seen application in experiences all over the world in recent years [51].

The phygital phenomenon can be interpreted as a radical change in the choices of tourists, at the individual and social level, who use devices as a direct source of information which affects their decisions and actions. The intelligent tourist destination, therefore, must now be considered as part of the tourist offer, with particular attention paid to those targets most sensitive to new technologies [43]. Within the tourist experience, however, the use of a phygital approach leads, compared to an exclusive use of digital technologies, to the perception of greater authenticity [52], which is sought by cultural and tourism businesses given that it positively influences tourists' visit experience [53]. In the very communication of cultural heritage companies, the intuitiveness of the phygital device affects how the heritage is perceived, and therefore the visit itself [54].

### 3. Methodology

To explore the promising link between cultural tourism startups and the phygital approach revealed by the literature review, the present study aims to investigate how cultural tourism startups exploit opportunities derived from the phygital formula, reporting practical cases of Italian startups that manage to top the market and enter a development phase, using this approach to generate new value propositions. This observation being

made in the field of startups is very important for describing the potential of the phygital formula, given the high reactivity and sensitivity of startups, both for better and for worse, to suggest the adoption of innovative approaches on the basis of their value proposition.

In this paper, the authors adopt a qualitative methodology [55] used multiple case study approaches [56], given the exploratory nature of the topic.

After an initial framing of the context, carried out by an exploration of the web based on secondary sources [57], such as reports, interviews, and reference consulting sites in the field of startups (Startupitalia.eu; Wired.it; Economyup.it accessed on 1 December 2023), the authors proceeded to define four case studies, namely, Loquis, Miravilius, Way Experience, and Mappina. For the identification of the latter, the 'think aloud' criterion, used in various studies on entrepreneurship, was chosen [58]. Within the online material that has been collected and analysed, there are secondary interviews and statements from the founders or managers of the selected startups, who respond with the 'think aloud' method to the questions posed to them and thus make it possible to obtain in-depth information on the opportunities that innovative enterprises can exploit thanks to the phygital approach, as well as the benefits that the phygital formula can bring to the stakeholders involved.

The evidence obtained was then organized through the use of an elaboration of the Value Triangle (VT) framework [59], which allowed us to understand the new value propositions generated by the startups in question due to their use of the phygital approach. In particular, at the core of the model is the value proposition, which can be defined as an explicit promise made by the company to its customers to provide a particular package of value-creating benefits [59,60]. Considering a company's commitment to creating value not only for itself and its customers but also for its partners and for society as a whole, the VT framework offers a broader perspective, allowing us to highlight how value is generated for all types of stakeholders (including the environment and future generations) as shown in Figure 1. Customer value represents the customers' perception of value as a trade-off between the benefits and sacrifices they bear in a specific usage situation. In order to understand the evaluation that customers made of the services offered by startups, the authors reported the reviews made on social media (Facebook) and, where specified, the corresponding rating.

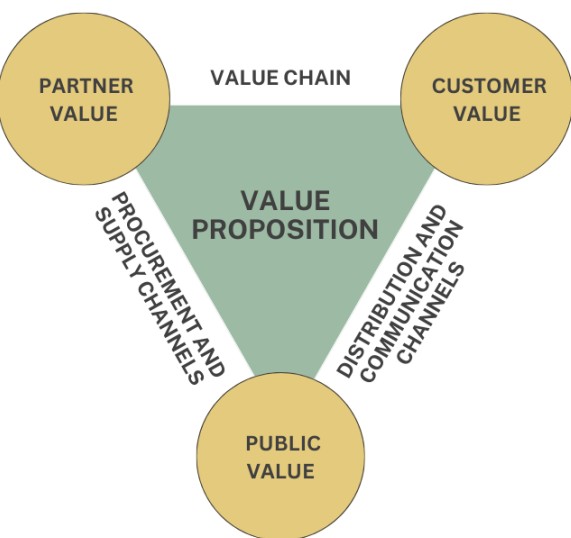

**Figure 1.** The Value Triangle framework for the value propositions generated by the adoption of the phygital approach. Source: Authors re-elaboration reused from Biloslavo et al., 2018 ([59] p. 755).

## 4. Findings

An increasing number of innovative companies in the field of cultural tourism and cultural heritage enrichment have experimented with new hybrid (online and offline) approaches. These approaches have enabled them to continue offering events and ex-

periences, suggesting that the COVID-19 emergency may have accelerated an essential process in the providing integrated services to tourists. This context has provided fertile ground for examining how Italian cultural startups in the tourism sector leverage phygital opportunities. In this research, the authors present the findings from the six selected case studies that demonstrate the utilization of phygital strategies by Italian cultural tourism startups. These cases are analysed and described through the framework outlined in the methodology section. Our analysis of the application of the phygital formula is conducted through the examination of three key variables: partner value, customer value, and public value. Each case study is scrutinized to understand how these startups have harnessed phygital opportunities to enhance their operations and value propositions. Furthermore, the impact of these phygital strategies on customers is explored. The results are presented in a structured manner, adhering to the defined framework, which allows for a comprehensive understanding of the startups' activities and their efforts to exploit the potential of the phygital approach.

In order to achieve categorisation of the analysed information, the following summary table (Table 1), illustrating the six analysed case studies, is proposed.

**Table 1.** Key information about the case studies.

| Startup Name | Value Proposition | Phygital Formula |
|---|---|---|
| Loquis | Enhancing cultural tourism with immersive, location-based storytelling through travel podcasts | Blending digital narratives with physical exploration |
| Meravilius | Virtual travel platform offering immersive, interactive global tours remotely | Integrating live digital tours with physical exploration |
| Way Experience | VR-enhanced cultural exploration blending history with digital immersion | Merging VR with tangible historical experiences |
| Mappina | Collaborative mapping platform enriching urban narratives through citizen engagement | Blending digital mapping with real-world insights |

Source: Authors' elaboration.

## 5. Implications

In relation to the evidence emerging from the analysis of the case studies (liked showed in the Boxes 1–4, this study provides an interesting picture of the different value propositions generated by applications of the phygital formula in the cultural tourism sector, with a focus on startups oriented towards the development of innovative solutions and predisposed to the use of new technologies.

From a theoretical point of view, it seems relevant to emphasize that most of the studies in the literature have focused on the users' perspective, considering the phygital approach a tool able to guarantee greater interactivity and customer involvement.

The use of the Value Triangle framework has made it possible to highlight that phygital-driven value propositions are numerous and involve a large number of stakeholders, fostering value co-creation processes in which users and cultural proponents assume almost (or fully) equal importance [61].

This analysis also provides interesting empirical evidence from which to derive greater awareness of the phenomenon and pave the way for advanced studies.

From a managerial point of view, this study highlights how the fusion of the physical and digital environment is highly relevant for the tourism and cultural sector, and, thanks to the analysis of the three key variables that emerged from the results, how it is possible to obtain a detailed picture of their different value propositions.

**Box 1.** Text. Example 1 about cultural-tourism start-up.

---

LOQUIS

*Value Proposition:*

Loquis is a pioneering app that revolutionizes cultural tourism through its "Travel Podcasting" platform. This digital tool offers an immersive experience by providing geolocated short travel podcasts linked to specific places. Its uniqueness lies in blending digital storytelling with physical locations, allowing users to explore areas through engaging narratives, without the need to constantly engage with their screens. The podcasts feature stories, anecdotes, and information narrated by locals or experienced travellers, enriching the visitor's experience with deeper insights into the visited areas.

*Partner Value:*

The Loquis app presents significant value to tourism bodies and local attractions. It serves as an effective platform for these entities to create and disseminate audio content that showcases their areas, providing tourists with informative and engaging guides. By utilizing Loquis, partners can freely promote their events and tourism offerings, leveraging the app's reach and interactive format to enhance visitor engagement and attract a wider audience to their locales.

*Customer Value:*

For its users, Loquis offers a particular way to explore and understand various locales. The app's user-friendly interface allows individuals to discover and listen to stories about their surroundings, whether on-site or remotely from home. This approach to travel enriches the user's experience by providing a diverse range of perspectives and narratives, enabling a more profound and personalized connection with the places they visit or aspire to explore.

The customer evaluation of Loquis through Facebook reviews is 4.6 out of 5. An emblematic quote that reflects its value proposition is the following: *'It gives you the opportunity to travel with the mind or learn anecdotes and curiosities about the places you are visiting.'*

*Public Value:*

Loquis contributes to public value by facilitating cultural exchange and enhancing people's understanding of different places. It democratizes storytelling by allowing anyone to create and share their experiences and perspectives on the platform. This inclusive approach fosters a diverse and multifaceted portrayal of locations, challenging stereotypes and providing a more comprehensive view of each site's cultural and historical significance.

Source: *Interview with the founder during Panel Webinar: 'PHYGITAL TOUR', digital experiences for the innovation of cultural tourism. (turismomusicale.net).* Accessed on 1 December 2023.

---

**Box 2.** Example 2 about cultural-tourism start-up.

---

MIRAVILIUS

*Value Proposition:*

Miravilius introduces an innovative approach to travel with its live streaming platform, offering virtual tours guided by professional tour guides from around the world. This "alternative travel" option primarily targets individuals unable to travel physically, allowing them to explore global destinations from home. The platform's unique feature is its interactive live tours, which provide a real-time, immersive experience, effectively changing the paradigm of travel by merging digital convenience with authentic exploration.

*Partner Value:*

The Miravilius platform significantly benefits tour guides and the tourism sector. By focusing on both Italian and international guides, especially in areas heavily reliant on tourism, Miravilius offers employment opportunities and helps sustain the industry. The platform keeps global interest in travel destinations alive, supporting tour guides by providing them with a digital platform to showcase their expertise and engage with a broader audience.

---

**Box 2.** *Cont.*

---

*Customer Value:*

For customers, Miravilius offers a novel way to experience travel. Users can embark on digital journeys, previewing destinations before physical visits. The platform caters to a new travel need: to explore and understand a place virtually to decide if it aligns with personal preferences. This digital exploration not only complements traditional travel but also encourages it, enhancing customers' decision making and planning processes for future trips.

An emblematic quote from the Volunteers Foundation, who used the service, is as follows: "it allowed 25 teenagers from the slums of Nairobi to live the exciting experience of the Live Tour of Cape Town, South Africa".

*Public Value:*

Miravilius contributes to the public good by making travel accessible to those who are physically unable to visit destinations, such as seniors and teenagers. The platform's social impact projects extend the joy and educational value of travel to a broader audience, fostering cultural understanding and curiosity. By providing virtual tours, Miravilius democratizes access to global cultural and historical sites, contributing to a more inclusive and informed society.

Source: *Interview with the founder during Panel Webinar: 'PHYGITAL TOUR', digital experiences for the innovation of cultural tourism. (turismomusicale.net).* Accessed on 1 December 2023.

---

**Box 3.** Example 3 about cultural-tourism start-up.

---

WAY EXPERIENCE

*Value Proposition:*

Way S.r.l. has established itself as an innovator in the digital cultural space by creating the first VR Tour, "You Are Leo", offering an unparalleled blend of tourism and virtual reality. Since 2019, this Milan-based startup has specialized in crafting unique phygital and fully digital experiences for the tourism and cultural market. Their goal is to ideate, produce, and disseminate projects that transform classical guided tours into rich, temporal, experiential journeys through augmented reality (AR), enhancing travel experiences.

*Partner Value:*

The strategic activities of Way S.r.l. are underpinned by a communication strategy that includes brand development, influencer marketing, and social media engagement. Their innovative approach has earned them recognition and awards, such as the ANGI prize for the most innovative company in Culture and Tourism in 2022. This acclaim has positioned Way S.r.l. as a valuable partner for cultural and touristic organizations looking to work with cutting-edge VR and AR technologies.

*Customer Value:*

Way S.r.l. offers customers immersive experiences that redefine the exploration of art and history. Through VR viewers, individuals can traverse time and space to experience accurate 3D recreations of historical settings and narratives. This allows for personal enrichment and entertainment, as well as a new form of engagement with cultural heritage, particularly appealing to younger audiences and fostering a deeper appreciation of the arts.

An emblematic quote from Facebook reviews that encapsulates its value proposition is *"A truly amazing and exciting immersive tour! It felt as if we were really there."*

*Public Value:*

The startup's impact on cultural valorisation is significant, supported by the Social Venture Foundation's impact4art project. Way S.r.l.'s vision—to spread culture, knowledge, and beauty through thrilling experiences—contributes to the public good by allowing for complete immersion in the art world. Their efforts aim to expand the audience for cultural works and sites, making cultural heritage more accessible and engaging for a broader community.

Source: the official website of the Way Experience company. Accessed on 2 December 2023.

---

**Box 4.** Example 4 about cultural-tourism start-up.

---

MAPPINA

*Value Proposition:*

MappiNA, originating from Naples, is an innovative collaborative mapping project that leverages mapping technologies to create a collective, culturally rich map, constructed directly by citizens. This thematic social network allows for the development of an urban narrative by capturing the city's diverse languages through text, photos, videos, and audio. It offers a platform for citizens to share their unique perspectives, including reporting cultural events and activities and highlighting lesser-known areas worth exploring. This approach provides a more comprehensive and multifaceted view of the city, moving beyond stereotypes and traditional representations.

*Partner Value:*

MappiNA's collaborative framework extends beyond individual contributors to include various stakeholders and partners. The project invites participation from residents, students, and cultural operators, encouraging them to contribute through diverse media. This inclusive approach enriches the map with these multifaceted perspectives, reflecting the real-life experiences and narratives of those who inhabit the city. By integrating inputs from various community members, MappiNA fosters collective effort in reimagining and reshaping urban narratives, thereby enhancing the cultural and social value of urban spaces.

*Customer Value:*

For users, MappiNA offers an engaging way to interact with and understand the cityscape. By allowing users to contribute to and access diverse content, the platform facilitates a deeper connection with the city's cultural and historical dimensions. Users can discover new areas, participate in cultural activities, and engage with the city in a more meaningful way. This interactive and participatory model enhances the user's experience of the city, offering a rich, layered understanding of urban spaces beyond conventional tourist routes and narratives. The customer evaluation of MappiNA through Facebook reviews is 4.9 out of 5. An emblematic quote that reflects its value proposition is as follows: *"Through the eyes of the users/visitors, we can discover curiosities, art, landscapes, and beauties of which we are sometimes unaware of their existence."*

*Public Value:*

MappiNA contributes to public value by promoting community engagement and cultural awareness. The project encourages citizens to actively participate in shaping the narrative of their city, fostering a sense of ownership and connection. By highlighting diverse aspects of urban life, including underrepresented areas and cultural activities, MappiNA enhances the public understanding and appreciation of the city's cultural richness. This collective mapping initiative also serves as a model for other cities, demonstrating the potential of collaborative technology to enrich urban life and cultural heritage.

Source: interview with the founder of the startup "Mappina" on Fanpage.it. Accessed on 4 December 2023.

---

The desire to explore the new frontiers of technology and to develop innovation-driven projects has led the cultural and tourism sector to enter into new partnerships and agreements with tech companies, who can provide expertise in the development of phygital solutions. In this way, the partner value is based on dynamic processes of collaboration between different actors operating in the same territory, who are able to encourage innovation and enter strategic partnerships for the communication and enhancement of cultural heritage.

From the perspective of the public and the community, the implementation of a phygital approach enables greater access to and knowledge of the world's artistic heritage, paving the way for the democratization of art, thanks to tools such as NFTs [62], and for more sustainable development [63], which can also preserve monuments and sites that are at risk due to serious international crises or excessive tourist flows. Several of the analysed case studies also show how the use of heritage-related hybrid technologies can foster the participation of the surrounding community in value co-creation processes, demonstrating the potential of collaborative technology to enrich urban life.

What we must never lose sight of is the value for the consumer, who, having no more limits to their fulfilment, can enjoy an enriched experience and broader involvement. The use of a phygital approach increases the attractiveness of cultural heritage to younger generations, who are interested in new and interactive experiences, and provides a privileged tool to increase accessibility for vulnerable audiences.

The phygital experience therefore becomes relevant also for the researchers themselves, allowing them, through a hybrid physical and digital modality, to take advantage of goods that cannot be used by all scholars due to physical limitations.

## 6. Conclusions

The analysis of these case studies highlighted the important contribution of innovative phygital models from startups to the cultural tourism sector. This is an approach that goes beyond the conception of technology as an element disconnected from reality, involving the senses to create real experiences that are enriched with new meaning, while also increasing their accessibility. The pandemic has had a key role in accelerating these processes. Furthermore, in the management of cultural tourism businesses, the organizational management aspect is fundamental, because often the moment of service provision coincides with that of its use by the consumer. Just think of the organization of guided tours, but also of cultural and musical events, exhibitions, theatrical shows, and concerts. The phygital approach now opens new horizons for experimentation, but, at the same time, it also requires new know-how, creativity, new skills, and professional figures; elements that find fertile ground in innovative ecosystems such as those of startups. In particular, as can be seen from our case studies, interactivity and the ability to personalize an experience are determining factors. Furthermore, alongside the sale of advertising space, online merchandising, and the offering of paid images comes the idea of using digital technologies as the basis for integrated subscriptions that include live experiences and remote use, elements that tourists are separated from after the moment of their normal visit, transform and extend the experience both before and after it occurs. This has a large impact on the tourist sustainability of places, but also, in many cases, has an extremely positive impact on the protection of places affected by excessive tourism, which are therefore not very accessible. It therefore emerges that studies exclusively on the public are important, although they offer a limited view: the phygital approach involves different subjects in the co-creation process, and therefore makes it necessary to place greater emphasis on the contribution of all stakeholders to these value-creation processes to also fully grasp the innovation processes that this type of approach brings to startups. Thanks to this approach, the possibility of the birth of value-creation communities is highlighted, where a key role is given to the younger generations. In the future it is, therefore, probable that research will focus more and more on models of cultural tourism startups and on how they will try to increasingly optimize their approaches by involving the public and all stakeholders, but above all future research should focus on the relationship between on-site visits and remote experiences.

**Author Contributions:** Conceptualization, F.G. and F.C.; methodology, F.B. and L.C.; Data Curation, F.G.; investigation, L.C.; Visualization F.C.; Supervision F.B.; Project administration F.B.; Funding acquisition, F.G. All authors have read and agreed to the published version of the manuscript.

**Funding:** This research was funded by by the FRA project (University Research Funding) of which one of the authors is the proponent and responsible. The code number of the grant is: P412.

**Institutional Review Board Statement:** Not applicable.

**Informed Consent Statement:** Not applicable.

**Data Availability Statement:** This paper presents exploratory descriptive findings from corporate case studies, highlighting the data access dates and links to the websites where the information was gathered.

**Conflicts of Interest:** The authors declare no conflict of interest.

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
