# Peer review of "Phygital as a Lever for Value Propositions in Italian Cultural Tourism Startups"

_sustainability, doi:10.3390/su16062550_

Round 1

Reviewer 1 Report

Comments and Suggestions for Authors

Dear Authors,

The paper has been written in a nice way. I am not sure 

if the citations are meant to be this way, but I cant click on any of them. That must be fixed.

Overall the paper concerns itself with 

The Rise of Phygital Cultural Tourism: How Startups Are Revolutionizing the Industry

The Pandemic Imperative: A Catalyst for Innovation

Case Studies: Unveiling the Power of Phygital Strategies

  • Offer theoretical contributions to the understanding of phygital applications in the cultural domain.
  • Provide practical recommendations for managers and founders of tourism-cultural startups as they navigate new business models.

A Focus on Enhancing the Visitor Experience

The selected startups primarily focus on:

  •  

This study highlights the transformative potential of "phygital" approaches in the cultural tourism sector. By embracing innovation and fostering collaboration between startups, cultural institutions, and policymakers, the authors can create a more engaging, accessible, and sustainable future for cultural tourism experiences.

The paper has a good structure.

Overall the paper is interesting and will be useful to future readers.

Author Response

Dear reviewer, thank you for your attention. We agree with your comments and assessments. We have proceeded to present the findings of our research more clearly and to expand upon certain key points in the literature.

Reviewer 2 Report

Comments and Suggestions for Authors

Regarding new tourism possibilities using current technology, the paper carefully describes several attempts by start-up companies and their advantages, rather than on a demonstration level. It is understood that there are many advantages, especially regarding cultural material, in terms of preservation, knowledge and image transmission, and assistance in human shortage. The usefulness of Phygital technology is mentioned, citing previous literatures.

However, there are lacks of objective measures such as user evaluation, which is essential for tourism, and visualization of the improvement in convenience through numerical values. Future research is expected. The paper also needs to include more information on how sustainability is linked to the use of each elements of Phygital technology and what elements are not transitory for tourism, and the rationale for this.

The followings are specifics on points to be considered.

Perceptions of tourism itself

(1) Tourism is an extraordinary experience, and therefore the technical elements required are not necessarily the same as those required in everyday life. The stakeholders are specified in figure 1, however, what is the “value” described in each of the items of the framework in figure 1? The authors have to clarify the elements of “value” in tourism.

<P2 Literature review>

Line5-8 "The contemporary literature delineates the phygital experience through the prisms of immediacy, immersion, and interaction, advocating for a seamless blend of physical and digital realms to create value- added consumer experiences (Taheri et al., 2019; Lupetti et al., 2015 )"

Line14 (2nd series beginning) "In dissecting the phygital approach, marketers identify three salient features-immediacy, immersion, and interaction."       

Some tourism that takes time and effort to travel to distant places. What aspects of immediacy are taken up in tourism? ã€€Elements required in everyday life are not always needed in an extraordinary and enjoyable trip.

(2) Description of current issues related to tourism

The word 'innovative' is used repeatedly, but its effectiveness will not stand out unless specific reference is made to current issues and how these issues can be solved through the uses of these technologies. It is not very effective if there is no specific reference to current issues and how these issues can be solved by using these technologies.

Lacks of valuations by stakeholders

The last line of the abstract states "improving customer interaction".

Is the customer the tourists or business partners in tourist destinations? The text does not seem to contain any specifics regarding customer (tourists) requests and improvements, nor does it seem to mention the results of those improvements. 

Customer value and partner value, shown in Figure 1, are specifically mentioned in several of the applications mentioned, but the basis for their mention is not provided.

 Ex. Pageï¼— “Loquis”

    "For the users, Loquis offers a unique and convenient way to explore and understand various locales... The app's user-friendly interface ..." 

If it is not an advertising statement, there needs to be evidences of what is “unique” and what is “user- friendly interface”. It is necessary whether this is backed up by user evaluations.

This is just one example, and although the author's high evaluations are mentioned in relation to other applications, the rationale and objective validity is lacking.

How exactly has customer convenience been improved? 

On sustainability

(1) Do many VR users actually go to tourist destinations?

The economic impact of real tourism is significant not only for tourist facilities but also for the entire region, including accommodation, transport, shops, etc. VR could help solve over tourism challenges. However, if VR is used, the sustainability of the business is less convincing if there is no reference to the method used to lead to actual visits.

(2) With regard to the likelihood of repeat visits by tourists.

Do such technologies generate repeat visitors who want to experience the same thing more than once? Positive references to sustainability are also required to be added, along with evidence

A spelling mistake.

Introduction Third series Line 10 post-Covida 19 →Line 10 post-COVID 19

Author Response

Dear reviewer, we are immensely grateful for your thorough and insightful review. We find ourselves in strong agreement with your observations and have endeavored to adjust the manuscript in accordance with your requests. Below, we succinctly summarize the adjustments made in response to your specific suggestions.

1) We have elaborated on our objectives in defining value for customers, also making reference to the framework you pointed out, as well as engaging with relevant literature on the subject. This clarification aims to deepen our discussion by integrating theoretical insights with practical implications, thereby enriching our understanding of customer value and its significance within the broader context of our study.

2) In response to your comment, we have endeavored to clarify the connection between our innovative approach and the utilization of hybrid physical and digital technological innovation. We acknowledge the importance of specifically addressing current issues and have thus aimed to demonstrate how these technologies can provide effective solutions. Our revised sections aim to make this link more explicit, illustrating the practical implications of these innovative technologies in addressing contemporary challenges.

In response to your comment on the lack of valuations, we have addressed this by incorporating into our presentation of findings, and consequently into the case studies, the consumer perspective through the use of numerical ratings and reviews expressed by consumers on social media platforms. This approach has been adopted to more effectively highlight the concept of customer valuation of the value proposition offered by the startups discussed. This addition aims to enrich our analysis by providing a tangible measure of how consumers perceive and value these startups' offerings, thereby offering a more nuanced understanding of their market impact.

2) (for lack of valuation) In our revised manuscript, we have made the decision to exclude case studies where we were unable to procure both numerical and descriptive evaluations from customers that could illuminate consumer valuations of the business cases in question. Specifically, we have removed two case studies from the previous version. In lieu of these, we have enriched the remaining case studies by providing detailed information on satisfaction ratings and incorporating reinforcing quotes. This adjustment allows us to present a more focused and substantiated analysis, ensuring that each case study included offers meaningful insights into consumer perceptions and the value proposition of the businesses under examination. This refinement strategy enhances the overall quality and relevance of our research findings.

Economic impact 1-2
Regarding the measurement of economic impact, we wish to convey that our paper retains a descriptive and exploratory nature, and thus remains qualitative in its approach. For a future extension of our research, we would certainly consider an expansion towards a more empirical measurement of performance. This potential direction acknowledges the importance of quantitatively assessing economic impacts while maintaining the foundational qualitative insights that underpin our current investigation.

We hope to have been thorough and precise in our responses, and we are immensely grateful for your valuable comments.

Round 2

Reviewer 2 Report

Comments and Suggestions for Authors

I have read the revised paper and appreciate the additions on Customer Value, including the reference to the customer evaluation. I hope that the evaluation axis will be made more concrete, and that objective and valid evaluations will continue. as clarifying the basis for the evaluations will make it easier to create future apps.